# Determination of Key Technical Parameters in the Study of New Pressure Sealing Technology for Coal Seam Gas Extraction

**DOI:** 10.3390/ijerph19094968

**Published:** 2022-04-19

**Authors:** Zhongguang Sun, Xuelong Li, Kequan Wang, Fakai Wang, Deyou Chen, Zhen Li

**Affiliations:** 1State Key Laboratory of the Gas Disaster Detecting, Preventing and Emergency Controlling, Chongqing 400037, China; sunzhongguang126@126.com (Z.S.); kingcasey@163.com (K.W.); 2College of Energy and Mining Engineering, Shandong University of Science and Technology, Qingdao 266590, China; chendeyou2021@163.com (D.C.); lizhen@sdust.edu.cn (Z.L.); 3China Coal Technology and Engineering Group Chongqing Research Institute, Chongqing 400039, China; 4State Key Laboratory of Coal Mine Disaster Dynamics and Control, College of Resources and Environmental Science, Chongqing University, Chongqing 400044, China; 5College of Mining Engineering, Guizhou Institute of Technology, Guiyang 550000, China; wangfakai316@126.com

**Keywords:** coal seam, gas drainage, disturbance crack, pressure sealing, polyurethane, cement slurry

## Abstract

Coal is affected by the concentrated stress disturbance of mining, the disturbance of drilling hole formation, and the concentrated stress of coal shrinkage and splitting of gas desorption from the hole wall; these result in a large number of secondary cracks that collect and leak gas. As a result, it is difficult for the coal seam sealing process to meet engineering quality sealing requirements, which results in problems such as low gas concentration during the extraction process. In this paper, based on the analysis of coal pore and fissure characteristics, and in view of the current situation of gas drainage and sealing in this coal seam, combined with the existing grouting and sealing technology, it is proposed to use pressure grouting and sealing to realize the sealing of deep coal bodies in the hole wall. According to the field conditions, the experimental pressure sealing parameter index is as follows: theoretical sealing length *L*_1_ = 9.69 m, the sealing length *L*_2_ = 13.98 m is verified, and the final sealing length is determined to be 15 m; the sealing radius is determined to be 0.6 m; the cement slurry was prepared on site with a water: cement ratio of 2:1; *P_G_* = 0.43 MPa was calculated; the range of the slurry diffusion radius *R* was 93.4–176.6 cm; the grouting pressure was determined to be 0.516 MPa. Field application practice has proved that: (1) Under the same drilling parameters and sealing parameters, the gas drainage effect of drilling with pressure sealing is 2.3 times higher than that without pressure sealing; (2) Using traditional sealing technology for drilling holes, the gas extraction concentration is far lower than the sealing operation effect of using the pressure sealing process; (3) Reasonably extending the length of the gas extraction drilling and sealing is a basic guarantee for realizing a substantial increase in the gas extraction concentration; (4) Sealing with pressure leads to a reliable and stable hole process.

## 1. Introduction

As a porous medium, the inherent structure of coal is complex and variable. However, according to the connectivity of pore development, it can be divided into effective pores and isolated pores [1], as shown in Figure 1a. According to the distribution law of pores and the arrangement of the pores, the effective pores can also be divided into parallel pores and series pores, as shown in Figure 1b. The effective pores not only form the coal base adsorption–gas desorption collection space, but also constitute the communication channel between free gas and coal.

As a companion to coal, coalbed methane is an efficient and clean energy source [1,2]. The calorific value of pure methane is 35.9 MJ/m^3^, which is equivalent to the calorific value of 1.2 kg of standard coal. In addition to being an energy source, coalbed methane also poses a threat to coal mine production, and causes hundreds of deaths each year [3,4]. Taking coal seam gas out for use not only reduces the risk of coal production, but also obtains clean energy and reduces environmental pollution [5]. This has a triple benefit, to safety, energy, and the environment. In 2015, China’s gas extraction capacity was 18 billion m^3^, utilization was 8.6 × 10^9^ m^3^, and utilization rate was 47.8%. Of this, coal mine gas drainage was 13.6 × 10^9^ m^3^, and utilization rate was only 35.3%, which is equivalent to 8.8 × 10^9^ m^3^. The gas extracted from the coal mine is discharged into the atmosphere, resulting in a serious waste of resources. The global warming potential of methane is 25; therefore, this is equivalent to the emission of 80 × 10^9^ m^3^ of carbon dioxide, which causes a significant contribution to the greenhouse effect [6]. The main reason for the low utilization rate of underground gas is that the gas concentration of the extraction is low and the extraction effect is poor. An important factor affecting the gas drainage effect of coal seams is the negative pressure of extraction. Under the action of the gas pressure gradient generated by the negative pressure, the gas flows continuously through the crack to the borehole, thereby achieving the purpose of gas drainage. However, the negative pressure in the borehole is affected by the quality of the borehole seal.

The gas extraction concentration is an important indicator used to measure the effect of gas extraction. In the actual production process, it is found that the gas extraction concentration gradually decreases with the extension of the extraction time. This is of course due to the reduction in the gas content of the coal, but another important reason is that the poor sealing quality leads to gas leakage in the borehole [7].

Due to the existence of cracks around the borehole and the poor sealing quality, the air outside the borehole enters the borehole under the action of the pressure gradient, resulting in a decrease in the concentration of the extracted gas. After the mine gas drainage borehole is formed, the coal around the borehole is damaged, and the stress reduction zone (I, II), the stress concentration zone (III), and the original rock stress zone (IV) are successively generated in the vertical and radial directions of the borehole. The stress-reduced zone is annularly distributed in space [8]. After the borehole is drilled, until the stress reaches equilibrium again, and the fissure area along the radial direction of the borehole is the area of the air leakage ring of the borehole. The stress distribution of the coal around the borehole is shown in Figure 2.

Here, *σ_r_* is the tangential stress in the plastic zone, *σ_ψ_* is the radial stress in the plastic zone, *q* is the borehole support force, *ψ* is the internal friction angle, *r* is the distance from any point around the borehole to the borehole, *r*_0_ is the borehole radius, and *P*_0_ is the original rock stress.

In addition to the gas leakage from the cracks around the borehole, the gas leakage in the hole is formed through the voids in the sealing material under the action of the negative pumping pressure [6]. The quality of the sealing material is an important factor affecting the quality of the sealing, and improving the sealing performance of the sealing material can further improve the extraction effect. One of the reasons for air leakage is that the sealing material is not in close contact with the wall of the drilled hole [5]. The hole wall of the borehole will deform slightly under the influence of stress, and the tight fit between the sealing material and the hole wall provides conditions for air leakage, resulting in air leakage during the extraction process. When a near-horizontal, small-angle gas drainage borehole is sealed, it is impossible for the sealing material to complete the sealing between the hole wall and the drainage pipe due to its own gravity. In addition, the pores of the sealing material itself are relatively large, and the gaps inside connect to form leakage channel.

Therefore, gas drainage is the result of a combination of factors, and its effect depends not only on the formation and occurrence conditions of coal seam gas, but also on the quality of gas drainage engineering. The difficulty for the gas drainage drilling and sealing effect to meet the engineering requirements is an important reason that most mine gas drainage is not up to standard [9]. As we all know, the sealing effect is the key to coal seam gas drainage. At present, the main problems with drilling and sealing engineering are as follows: it is difficult for the quality of gas drainage and sealing of this coal seam to meet the requirements of drilling seal inspection; the negative pressure of extraction is low and the concentration of gas is too small. The analysis found that the current status of gas drainage in a coal seam is caused by the gas drainage and sealing process currently applied to the coal seam, for which is difficult to meet the technical requirements of drilling engineering under this condition. Therefore, there is an urgent need to propose a new type of coal seam gas drainage and sealing process to solve the technical deficiencies of the current coal seam gas drainage and sealing.

Conducting mine gas drainage and utilization does not only reduce the occurrence of coal mine gas accidents, but also purifies the atmospheric environment of the mining area [10]. Moreover, as high-quality energy, the utilization of gas can create enormous economic value for society [11,12]. The process of sealing and sealing boreholes is specially designed for the current coal seam gas drainage and it is difficult for sealing quality to meet the engineering technical requirements; therefore, a new coal seam drilling and sealing method is proposed. The process can significantly improve the sealing quality problems existing in the current coal seam gas drainage process: sealing the gas leakage channel; sealing the gas extraction gas chamber; increasing the gas drainage concentration; increasing the coal seam gas drainage rate and prolonging the extraction cycle; reducing the risk of coal and gas outburst in coal seams, reducing the cost of mine ventilation; and providing a prerequisite for the development and utilization of gas.

The objectives of this study are to determine the optimum sealing length of the pressurized hole sealing process; the optimum grouting and sealing radius; the optimum water:cement ratio of the slurry; the optimum grouting pressure; and the effective radius of slurry diffusion according to the specific conditions of the site.

## 2. Current Problems with Sealing Technology

The suction pressure at the bottom of the borehole has the effect of draining gas and forcing gas desorption [5]. The quality of the sealing borehole is directly related to the effect of gas drainage [8]. At present, the main problems with drilling and sealing engineering are as follows: the quality of gas drainage and sealing of the coal seam is difficult to meet the requirements of drilling seal inspection; the negative pressure of pumping is low; the concentration of gas is too small. At present, the sealing technology adopted at home and abroad mainly includes mechanical cement mortar sealing, foaming polymer sealing, and sealing device sealing [13]. Among these technologies, cement mortar sealing is mainly suited to inclined drilling [12], which is not suitable for near-horizontal or gently inclined coal seams; foamed polymeric material sealing has the advantages of high foaming ratio, light weight, and rapid sealing of polymeric materials, but the material cost of the borehole is high and the operation requirement is high; the fast sealing device has a high sealing speed, and repeated use can reduce the cost, but the effect is poor, and it is only suitable for temporary drainage and sealing [11]. In general, the existing gas drainage method is limited only to “block” the upper borehole; it does not involve blocking and dealing with coal seam fissures and gas leakage channels [9]. These coal seam fissures and gas leakage channels will develop and expand with the extraction of gas, resulting in gas potential in the later stage of pumping. It is released, causing the deformation, displacement, and pressure relief of the coal. Under the action of suction negative pressure, air can easily enter the borehole through these cracks, resulting in a decrease in gas drainage concentration and shortening the drainage cycle of gas drilling.

## 3. Experimental Study on Pressure Sealing Borehole in Coal Seam

### 3.1. Principle and Procedure

Coal has a porous nature with a large internal surface area [14]. When undisturbed by the project, the structural surface is in a tightly squeezed state, but the degree of bonding is poor. When disturbed by the project, on the one hand, it will cause certain mechanical damage to the gas-containing coal in contact with the coal wall; on the other hand, it breaks the original stress balance state of the coal, causing disturbance and damage to the coal within a certain range of engineered contact [15]. In addition, the coal adsorbs high-pressure gas, and the engineering disturbance increases the gas pressure gradient of the coal, causing a large amount of coal gas to be desorbed and released, and the coal shrinks and deforms, further expanding the coal crack around the borehole; the closer to the borehole center, the more disturbing the crack is [16,17]. These disturbance fissures constitute the main channel for gas desorption and release in the initial stage of the extraction drilling [18,19]. With the extension of time, the fissure extends to the coal around the borehole, forming a channel for gas leakage and air communication.

The pressure sealing borehole technology uses the grouting equipment to inject the slurry material into the gas sealing borehole sealing section space and the surrounding coal wall disturbing crack inside the borehole wall with a certain pressure. Under the action of grouting pressure, the slurry can split and expand the coal cracks in the pore wall, filling the pores and the coal surface [20,21], increasing the slurry diffusion range, and infiltrate microcracks under the large osmotic pressure gradient. The cohesive force is generated; after the slurry is solidified, a dendritic distribution is formed and bonded to the coal particle solids to seal the gas leakage passage.

According to the slurry permeation mechanism, the slurry flows, permeating and diffusing along the crack under a certain grouting pressure gradient. When leaving the grouting pipe at a certain distance and the pressure gradient is reduced to the critical pressure gradient, the slurry flow rate decreases and the turbulent flow changes to a laminar flow state. The hydraulic material will solidify at this time; the viscosity increases and the flow rate continues to decrease. Finally, slurry stops flowing. Particles of non-hard materials gradually coalesce and precipitate or adhere to the cracks, which increases the flow resistance and static shear stress of the slurry [22]. In addition, column grouting is formed at a certain distance inside the front end of the grouting pipe and the surrounding coal. The slurry forms a network skeleton on the fracture surface [20,23] and fills into the fine cracks of the fractured rock mass. After solidification, it forms an organic plastomer with high cohesiveness to the coal, which improves the fracture environment of the surrounding broken coal, and then meets the drilling sealing technical requirements [24].

After pressure grouting, slurry solidification can significantly increase the strength of the coal [25]. Figure 3 shows the electron microscopy scan of the Luling Coal Mine of Huaibei Mining Bureau after high-pressure injection of coal samples [26]: (1) at the fractures and pore fractures [27], it can be seen that after the slurry is consolidated, the particles are filled and the fracture is closed, which effectively blocks the pores; (2) the slurry undergoes osmotic solidification under the action of pressure, and has a trunk and branches [28] that are integrated into one body and bonded with coal particles to form a huge cohesive force and increase the strength of the coal [29].

Through the above analysis, it can be concluded that the use of pressure grouting can completely block the cracks in the borehole wall, strengthen the coal, and avoid secondary borehole cracks.

When polyurethane is used to fix the grouting sleeve, the polyurethane is required to withstand a certain grouting pressure. According to this, the space of the drilling and sealing borehole is closed before and after using the polyurethane, and then the grouting device is used to inject the cement slurry into the sealing section space through the grouting pipeline.

Before sealing the borehole, use the compressed air to blow the drill cuttings in the borehole. Then, the gas drainage pipe is placed at a certain depth in the drainage borehole, and the grouting space of the drainage borehole sealing section is quickly constructed using polyurethane. After that, the grouting pipe is reserved through the orifice end, and cement slurry of a certain pressure is injected into the closed space of the sealing section by the grouting pump. After the grouting is completed, when the cement is solidified, the water is pressed by the pressure air or the grouting pump to check the sealing effect of the drainage borehole. If the inspection is passed, the gas is connected to the grid. The sealing is shown in Figure 4.

According to the principles of sealing technology, the technology of pumping and sealing boreholes is a mechanical–manual hybrid sealing method. Among them, the manual operation part, that is, the polyurethane construction sealing section, grouts the closed space. Generally, the foaming time of the mining polyurethane is 2 to 3 min, and it is necessary to compare the manual operation time of the pressure sealing borehole. For this purpose, a set of simulated sealing operation drilling process is prepared. The results show that the process operation time is about 2.5 to 3 min. In addition, considering that the temperature of the mine is higher than the ground temperature, the chemical reaction rate will be accelerated. Therefore, a time-delay foamed polyurethane has been specially developed; the basic parameters are shown in Table 1.

### 3.2. Parameter Determination

#### 3.2.1. Grouting Slurry

Determine the proportion of cement slurry, so that it can achieve optimal penetration, plugging the cracks, joints, and isolation cracks in the coal around the pumping borehole. The test determines the basic properties of the water:cement ratio of the cement slurry. The test data are determined by using ordinary Portland cement No. 425. The test data are shown in Table 2.

#### 3.2.2. Sealing Length

In order to avoid the influence of roadway disturbance crack pressure on the sealing borehole sealing section and improve the sealing effect of coal seam gas drainage, the influence range of roadway disturbance cracks is mainly considered when designing the length of sealing section. When the credibility interval is 95%, the reasonable sealing length of gas drainage boreholes in the working coal face can be calculated by [30]:(1)$$L_{1}=2.63b_{1}\cdot F_{0}^{0.42}$$
where *L*_1_ is the length of the coal sealing section of the working face, m; *b*_1_ is the width of the coal roadway of the working face, m; then, $F_{0}$ is the time standard, dimensionless [30]:(2)$$F_{0}=4\lambda P_{0}^{1.5}t/(ab_{1}^{2})$$
where λ is the coal seam permeability coefficient, m^2^/MPa^2^·d; $P_{0}$ is the original gas pressure (absolute), MPa; t is working surface exposure time, d; and a is the coal seam gas content coefficient, m^3^/m^3^·MPa^1/2^.

Similarly, the formula for calculating the sealing length of the gas drainage borehole in the two-way coal seam of the roadway with a simulated gas pressure distribution interval of 95% is [30]:(3)$$L_{2}=4.25(\lambda P_{0}^{1.5}t/a)$$

Compare the results of the two parameters and take the larger data as the length of the borehole. Therefore, as long as the basic parameters are input into Equations (1) and (3) for calculation, the reasonable sealing length of the gas drainage borehole of the coal seam can be obtained.

#### 3.2.3. Grouting Radius

The grouting radius is the effective range achieved by the diffusion of the slurry, mainly based on the range of impact of the borehole disturbance. The grouting radius affects the compactness of the coal within the grouting reinforcement range. Under the condition of the grouting borehole arrangement, the larger the grouting diffusion radius, the higher the compactness of the coal filled with the slurry, and the better the grouting effect.

The grouting aperture is determined according to the pipe diameter and the bore diameter of the pipe, and the principle from the inside to the outside should be satisfied. Which is:(4)$$r_{1}<r_{2}+10\;\mathrm{mm}$$
(5)$$r_{3}+10\;\mathrm{mm}<r_{4}$$
where *r*_1_ is the borehole diameter of the extraction borehole, mm; *r*_4_ is the inner diameter of the extraction pipe, mm; *r*_2_ is the outer diameter of the extraction pipe, mm; and *r*_3_ is the aperture of the opening, mm.

The research indicates that the disturbance range by gas extraction is generally 5~10 times of the pore diameter. Combined with the pressure relief extension of coal fissure with time under the action of earth stress, it can be determined that the influence range of borehole disturbance crack is larger than the calculated value [31]. Therefore, the disturbance crack caused by the drilling activity is within the influence of the sealing radius.

#### 3.2.4. Grouting Pressure

Grouting pressure is the theoretical pressure of grouting required to achieve effective sealing radius when sealing with pressure. The flat grouting experimental shows the relationship between the diffusion radius and the grouting pressure, slurry viscosity, and grouting time [32]:(6)$$R=\sqrt[{2.21}]{\frac{0.093(P_{G}-P_{0})T\delta^{2}r_{0}^{0.21}}{\mu }+r_{0}}$$
where *R* is diffusion radius, cm; *P_G_* is grouting pressure, Pa; *P*_0_ is coal seam gas pressure, Pa; T is grouting time, min; μ is slurry viscosity, mPa·s; δ is crack or the width of the pores, cm; and $r_{0}$ is the radius of the borehole, cm.

In order to ensure the effect of grouting, combined with the theoretical theory of the above theory and the experience of on-site coal seam grouting technology, the shift in Equation (6) is obtained [31]:(7)$$P_{G}=\frac{\mu {\left(R-r_{0}\right)}^{2.21}}{0.093T\delta^{2}r_{0}^{0.21}}-P_{0}$$
where *P_G_* is grouting pressure, Pa; *P*_0_ is fissure water pressure, Pa; *T* is grouting time, generally about 15, here 900 s; *r*_0_ is drilling radius, due to the pore size of the coal borehole Uniformity, here taken as *r*_0_ = *ar*/2, *a* is the drilling unevenness coefficient, generally taken as 1.2, cm; *R* is sealing radius.

Next, in order to verify the accuracy of the obtained grouting pressure, the grouting pressure obtained by Equation (7) is brought into the Equation (8) [32] for inspection.
(8)$$R=283.82P^{0.53}M^{0.23}\mu^{-0.83}T^{0.55}$$
where *P* is grouting pressure (i.e., *P_G_*), Pa; *M* is size modulus of coal, k, *M* = 2.555 × 10^2^
*k*; μ is viscosity of cement slurry, mPa·s; *k* is coal permeability coefficient, cm/s; *T* is grouting time, min.

If Equation (8) results in a slurry diffusion radius greater than that of Equation (7), the grout radius can be proved to meet the theoretical requirements.

According to the experience of on-site grouting, in order to optimize the grouting effect, it is necessary to increase the pressure on the basis of theoretical calculation so that the slurry can be pressed into the coal crack better. Therefore, the actual grouting pressure is calculated as [30]:(9)$$P_{s}=\gamma P_{G}$$
where *P_s_* is actual grouting pressure, MPa; *P_G_* is theoretical calculation of grouting pressure, MPa; γ is redundancy factor, taken as 1.2.

When applying, the rationality of the grouting pressure is checked by the Equations (7) and (8), and Equation (9) is used as the on-site grouting pressure.

## 4. Field Application

### 4.1. Experimental Parameter

Pingmei Co., Ltd. (Pingdingshan, China) No. 10 Mine is located in the eastern part of Pingdingshan Coalfield. The location is shown in Figure 5. The minefield boundary is 4.8 km long from east to west and 7.5 km wide from north to south.

The results of mine gas grade identification over the years show that the absolute gas emission and relative gas emission are generally increasing, while the CO_2_ emission is basically constant. According to the safety production industry standard of the People’s Republic of China [33], the No. 10 Mine has been identified every year as a high gas mine every year, and the absolute gas emission of over mine is shown in Figure 6. It can be seen from Figure 6 that the difficulty of gas control in mines is increasing, the situation regarding gas control is severe, and there is an urgent need to improve the existing gas extraction process.

The 20180 mining face is located in the fourth stage of the Central Group of the No. 10 Mine. The mining face is located in the east of the mining area, three downhills, west to the east of the No. 25 exploration line, 200 m east, and the south is the 20,160 goaf, the north has not yet been mined; a diagram of the relationship between coal gas content and depth is shown in Figure 7. It can be seen from Figure 7 that the test data points and the fitting relationship curve in the figure have a linear relationship; that is, with the increase in the coal seam burial depth, the gas content increases linearly, which greatly increases the gas drainage time.

The upper Ding and the lower coal seams corresponding to the working face are not recovered, and the proximate analysis and coal type of group coal are shown in Table 3.

The elevation of 20,180 working face is −510~−570 m, the thickness of coal seam is 1.1–1.6 m (the coal seam histogram is shown in Figure 8), the coal seam inclination angle is 10°–14°, the effective strike length is 915 m, and the mining length is 200 m. The measured gas content of the mining face is 15 m^3^/t, the gas pressure is 1.2 MPa, and the gas permeability coefficient of the coal seam is about 0.005–0.08 mD.

Considering the unloading and fissure of the borehole, the drilling distance should be greater than twice the influence radius of the unloading when determining the drilling parameters. The test borehole diameter is Φ120 mm, which is calculated according to the radius of influence of the maximum borehole crack, that is, the two-borehole spacing should be greater than 1.2 m.

(1)Sealing length

According to the geological occurrence and geological occurrence of coal seams in 20180 mining face, the length of the borehole sealing calculated by Equations (1) and (2) is *L*_1_ = 9.69 m, and *L*_2_ = 13.98 m is verified by Equation (3). Considering the influence of a certain roadway disturbance crack, it is determined that the 20180 mining face coal seam gas drainage sealing length is 15 m.

(2)Sealing radius

The coal seam of 20,180 mining face gas drainage drilling borehole diameter is about Φ120 mm. Considering the influence of the deep extension of the pressure relief crack on the drainage borehole, consider taking 10 times the aperture according to the maximum disturbance damage range, that is, the maximum sealing radius of the drill borehole is determined to be 0.6 m.

(3)Sealing grouting pressure

In order to completely block the gas leakage crack channel, according to the test results, ordinary Portland cement No. 425 was selected, and the cement slurry was prepared on site by using a water:cement ratio of 2:1. The calculation parameters of Equations (7) and (8) are shown in Table 4.

According to the parameters of Table 1, Equation (7) is used to calculate *P_G_* = 0.43 MPa; using Equation(8), the slurry diffusion radius *R* is in the range of 93.4 to 176.6 cm, which is much larger than the empirical grouting sealing radius of 60 cm. Therefore, the grouting pressure satisfies the requirements. Finally, the grouting pressure calculated according to the Equation (9) was 0.516 MPa.

### 4.2. Experimental Results

The length of the conventional gas drainage drilling borehole is 6~10 m, and the initial gas volume fraction after sealing is 30~90%; after 20~40 d, the volume fraction is reduced to 3~9%. The drilling construction parameters are basically the same during the pressure sealing test, and the sealing difference is mainly in the sealing process. The negative pressure of the orifice is 10~16 kPa. After 2 months of test observation, 1# (unpressed grouting) and 2#–6# (pressure grouting is 0, 0.20, 0.27, 0.34, 0.40 MPa, respectively) drilling tests were performed. The comparison of drilling gas concentration in each time period is shown in Figure 9.

In Figure 9, the 1# borehole has adopted a traditional non-pressure sealing method; the sealing length is short, and the gas concentration is rapidly attenuated, indicating that the crack around the drilling borehole gradually develops under the negative pressure of the drilling and pumping, and the outside air enters the drill through the crack. The pores cause a drop in the concentration of the pump. When the pressure is too large in the 2# borehole with pressure grouting, the slurry opens the polyurethane. It can be considered that the pressure is 0, which is a certain increase with respect to the pumping concentration of the 1# borehole, but with an increase in the pumping time and the pumping of the negative pressure. The effect of the pumping concentration decays rapidly after 18 days. For 2# to 6# boreholes, wall coal crack sealing is more effective than the 2# pumping borehole with no pressure sealing borehole. The average gas drainage concentration of 1# borehole is only about 21.3% of the average concentration of 6# borehole, and the average gas drainage concentration of 2# borehole is only about 47.6% of the average concentration of 6# borehole, indicating that the pressure sealing process can improve gas drainage. The concentration is 30% to 55%, and the pumping time is extended by about 40 days.

It can be seen from Figure 10 that the gas concentration curve of 1# is significantly lower than the gas concentration of 2# to 4# borehole. Comparing the gas drainage concentration data in Table 5 and calculating the calculation, the average gas drainage concentration of 1# borehole is about 30.5%, and the average extraction concentration of 2# borehole gas is about 70.3%, which is the average gas drainage concentration of 1# borehole. Only the average concentration of 2# is about 43.3%. In combination with the on-site process, the 1# borehole pressure grouting section failed to achieve the pressure target, while the other operation links and material engineering dosages were the same as those of 2# to 4# boreholes for pressure sealing. Therefore, it can be concluded that under the same drilling parameters and sealing parameters, the effect of the pressure sealing is 2.3 times higher than that of the unsealed sealing.

Each gas extraction borehole pumping negative pressure changes with time, as shown in Figure 10. It can be seen from the figure that the pumping negative pressure fluctuations and fluctuations are consistent during each time period, so the influence of the pumping negative pressure of the drilling extraction on the gas drainage concentration can be eliminated.

The experimental results show that the cracking of the coal in the 2# to 4# borehole with the pressure sealing borehole is more effective than that of the 1# pumping borehole with the pressure sealing. In order to further explore the relationship between the average gas concentration value of each borehole of 2# to 4# boreholes and the pressure value of the pressure sealing borehole, the smooth curve scatter plot is visually presented, and the fitting curve is added, as shown in Figure 11. It can be concluded that the pressure sealing borehole can be used to improve the crack environment in the borehole wall around the gas drainage borehole. It is also stated that within a certain range of grouting pressure, as the sealing pressure increases, the sealing effect of the gas drainage borehole of the coal seam will be gradually improved.

## 5. Discussion

### 5.1. Influence of Coal Properties on Process

Coal fissures are divided into primary fissures and tectonic fissures. The primary fissure is formed during the coal formation process; the tectonic fissure is controlled by the regional tectonic stress field, which is mainly developed on both sides of the fault or at the junction of the structure and the stress concentration zone [34]. The coal adsorbs gas to store gas energy and generates adsorption expansion; absorbs geostress to store geological elastic potential, closes the spontaneous cracks in the coal and produces plastic deformation; the degree of microporous fracture development determines the gas permeability coefficient of the coal seam [35].

The pores in coal can be divided into: micropores, small pores, mesopores, macropores and visible pores and fissures [36]. Micropores: diameter < 10^−5^ mm, which constitute the adsorption volume in coal; small boreholes [37]: diameter = 10^−5^ to 10^−4^ mm, which is the space that constitutes capillary condensation and gas diffusion; mesopores: diameter = 10^−4^~10^−3^ mm, which is a laminar permeation interval that constitutes a slow gas; large pore: diameter = 10^−3^ to 10^−1^ mm, which is a strong laminar permeation interval and a plugging material percolation interval, and is determined to have a strong destruction of structural coal failure surface [38]; visible pores and fissures: diameter > 10^−1^ mm, which is the mixed permeation interval of gas laminar flow and turbulent flow, and determines the macroscopic failure surface of coal [39].

The crack width [40] (i.e., the opening degree of the fracture surface) is mainly caused by the tensile fracture of the coal caused by the tensile stress or the shear displacement of the structural plane. The size of the fracture width is usually closely related to the size and mechanical origin of the fracture surface. The larger the crack size, the larger the crack width. The tensile or tensile shear cracks are larger, while the compressive or compressive shear cracks are smaller.

The crack width mainly affects the grouting slurry ratio, grouting pressure, and slurry diffusion radius of the fractured rock mass [41]. For cement-based slurry and rheology-like grouting slurry similar to cement slurry, when the grouting pressure is constant, the crack width is large, and the slurry diffusion radius is also increased, the grouting slurry concentration can be increased. Conversely, the crack width is small and the slurry diffusion radius is small, and the concentration of the slurry should be reduced.

The state of the fracture surface includes the fracture surface connectivity and the crack filling condition [42]. There are very few cracks that are fully open, and most of the cracks have localized contact. The structural surface contact area includes a direct contact area and an indirect contact area between the two walls, the distribution of which is extremely complicated. Structural surface permeability is a function of the contact area of the structural surface, which decreases as the contact area increases.

Although the slurry sometimes fills the crack, there is a continuous water film between the slurry and the contact surface due to the presence of water droplets on the surface of the coal or the poor properties of the slurry [43]. Due to the presence of the water film, the coal and the slurry are isolated to greatly weaken the bonding force. Therefore, the presence of the water film will weaken the shear strength at the interface between the coal and the slurry, and the effect of grouting will be weakened. Since the coal is hydrophilic, the wetted phase liquid should be given priority in the process of grouting, and the wettability is better than that of water.

In addition to the wetting action, the slurry and the surface of the coal also have a bonding effect. This effect can be expressed by the adhesion force [44]. It can be seen that reducing the surface tension of the slurry can improve the wettability of the slurry and the coal, and facilitate the flow of the slurry, but the adhesion and affinity will decrease, thus impairing the bond strength between the slurry and the coal and the permeability of the slurry, thereby affecting the injectability and grouting effect of the slurry. In order not to impair these two effects, only the interfacial tension between the slurry and the coal is lowered.

The state of the fracture surface also has the undulating shape of the fracture surface, which also affects the grouting effect [45]. The undulating shape of the structural surface is generally non-flat, and has a certain curvature, generally divided into a stepped ridge, a wave shape, a straight line, a zigzag shape, etc [46,47]. The mean curve rate is the ratio of the actual process length of the slurry in the structural plane to the linear distance from the grouting borehole to the slurry flow to a point. When the slurry reaches the designed diffusion radius, the greater the curvature rate, the greater the power loss of the slurry processes, and the grouting pressure required is relatively high.

### 5.2. The Effect of Grouting on the Process

According to the definition of grouting reinforcement process, according to the distribution pattern of slurry, the injection method can be divided into: filling grouting, infiltration grouting, split grouting, splitting penetrating grouting [48]. In the coal cracks grouting, the common injection methods are filling grouting, penetrating grouting, and so on.

Filling grouting generally involves large boreholes and large pores, the purpose of which is to strengthen the entire coal seam to improve the stability of the coal. Therefore, the grouting materials commonly used are mainly suspended slurries such as cement slurries and cement water glass. Filling and grouting in the roadway often uses low-pressure grouting of cement slurry, so it is difficult for the slurry enter the microcracks of the coal, and its plugging effect is limited.

Infiltration grouting means that the slurry penetrates into the cracks or pores of the coal under a certain grouting pressure, but the permeability is different due to the different shape of the crack or pore [49,50,51]. The grouting material for the infiltration grouting of large pores is similar to filling grouting, and cement grout, cement clay grout, cement water glass, etc., can be used, and the stone body strength is high. For larger pores, a suspension such as cement slurry can be used. For medium porosity, the slurry is primarily selected based on the permeability of the coal and the stability of the gel. The choice of grouting pressure should be determined according to the permeability of the rock and soil, the time of slurry gelation, the amount of grouting, and the range of penetration. When the slurry injection time is long, the slurry generates a gel during the flow, which blocks the cracks or pores, thereby increasing the grouting pressure. Therefore, the correct choice of gel time and grouting method can achieve good grouting effect.

### 5.3. Practical Significance of Grouting and Sealing for Gas Drainage

Gas drainage drilling with pressure sealing technology research can meet the disturbance cracks around the drainage borehole, increase the gas drainage concentration, extend the gas drainage cycle, and increase the gas drainage rate.

The confirmed practical significance is:(1)Minimizing the possibility of mine gas disasters and improving coal mine production safety. In this study, a large amount of gas stored in the coal seam can be pumped out through the extraction drilling borehole to reduce the residual gas content in the coal seam and prevent the occurrence of gas disasters. At the same time, reduce the pressure on the ventilation system caused by the gas emission from the working face, and save ventilation costs. This plays an important role in guiding gas extraction work, reducing coal mine gas emission, and highlighting dangers.(2)Conducive to the development and utilization of coalbed methane resources, energy saving, and emission reduction. In order to comprehensively utilize gas resources and advocate sustainable development strategies, most high-gas mines have built gas-fired power generation facilities to meet the needs of daily life, and use the extracted coal seam gas to develop and create added value. This study can provide sufficient and secure sources of coalbed methane for gas power stations, and therefore has important social and economic benefits.

## 6. Conclusions

Through on-site investigation, laboratory simulations, and on-site operation verification, this paper analyzes the advantages and disadvantages of each sealing process for coal seam gas drainage drilling wells, combined with coal seam disturbance crack formation mechanisms, based on grouting technology theory and field experience. The pressure sealing technology of gas drainage in this coal seam is proposed.

(1)The principle of pressure sealing the coal seam gas drainage belt is proposed—that is, the pressure grouting is used to realize the coal gas release crack in the sealing borehole wall. The technology is based on the objective existence of concentrated stress disturbance cracks in the coal roadway of the working face. The pressure grouting is used to block the gas leakage micropores and crack channels. After the slurry solidifies, the organic plastomer with high adhesion to the coal is formed and improved. The coal wall fissure environment of the borehole wall improves the integrity and uniformity of the coal.(2)Through on-site inspection, the gas drainage effect is significantly improved. The theoretical sealing length L_1_ = 9.69 m of the Wu 9-20180 mining face is calculated; the sealing length L_2_ = 13.98 m is verified, and the final sealing length is determined to be 15 m; the sealing radius is determined to be 0.6 m; The cement slurry was prepared on site with a water:cement ratio of 2:1; P_G_ = 0.43 MPa was calculated; the range of the slurry diffusion radius R was 93.4–176.6 cm; the grouting pressure was determined to be 0.516 MPa. The on-site sealing experiment of the gas drainage hole shows that the pressure sealing process can increase the gas drainage concentration by 30% to 55%, and prolong the drainage time by about 40 days. The effect of sealing the hole with pressure is 2.3 times higher than that of the hole without pressure.(3)The process meets the technical requirements of the gas drainage and drilling engineering of coal seams, and provides a new scientific and effective sealing method for the direct extraction and utilization of coal seam gas. It is worth promoting.

## Figures and Tables

**Figure 1 ijerph-19-04968-f001:**
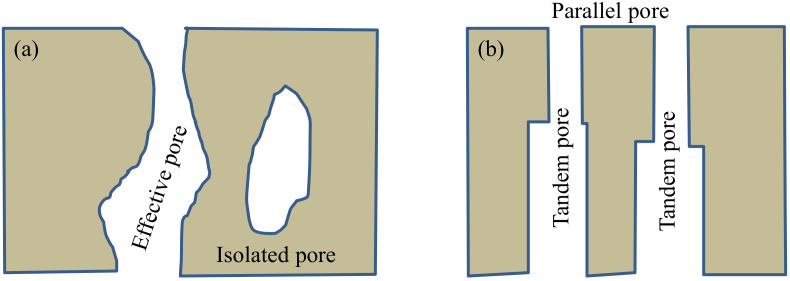
Diagram of coal pore type. (**a**) Schematic diagram of effective pores and isolated pores. (**b**) Classification of the effective pore.

**Figure 2 ijerph-19-04968-f002:**
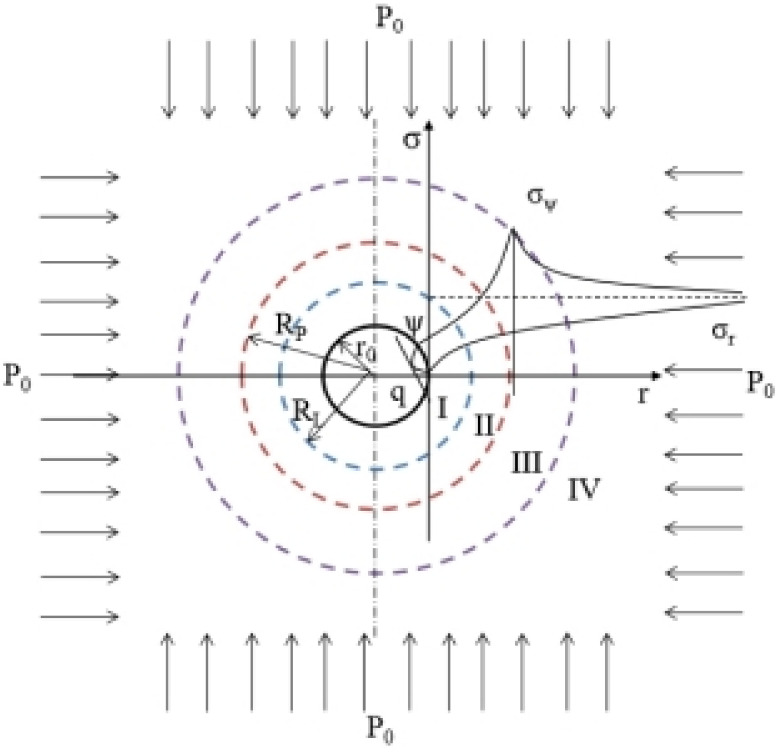
Stress distribution around the borehole.

**Figure 3 ijerph-19-04968-f003:**
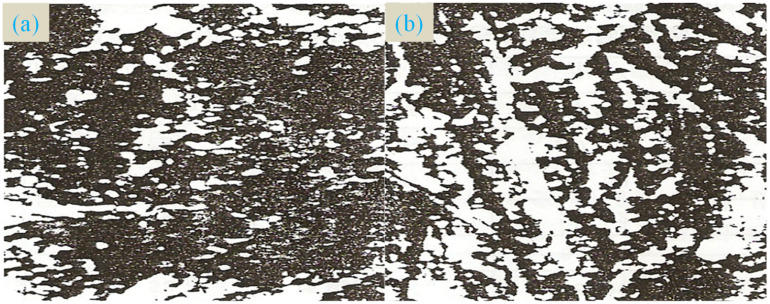
Electron microscopy of coal before and after pressure grouting: (**a**) before, (**b**) after.

**Figure 4 ijerph-19-04968-f004:**
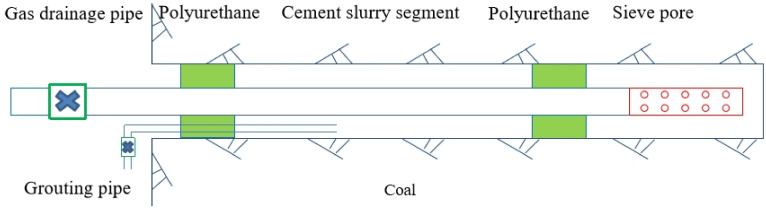
Schematic diagram of polyurethane and slurry sealing.

**Figure 5 ijerph-19-04968-f005:**
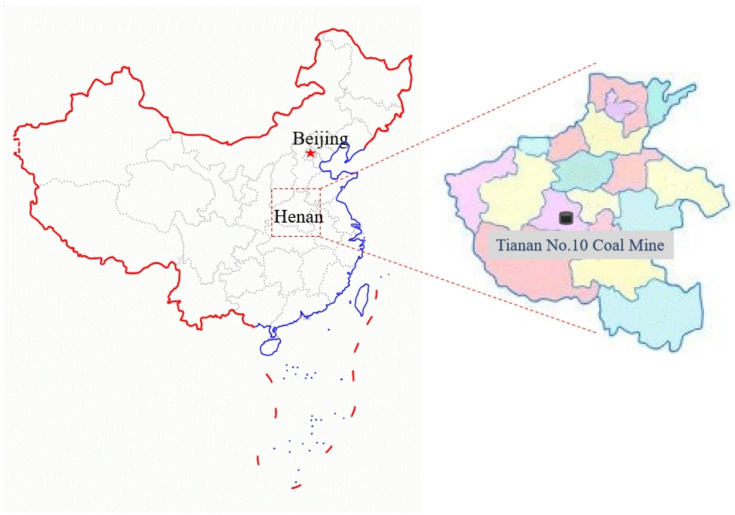
Location of No. 10 mining shares.

**Figure 6 ijerph-19-04968-f006:**
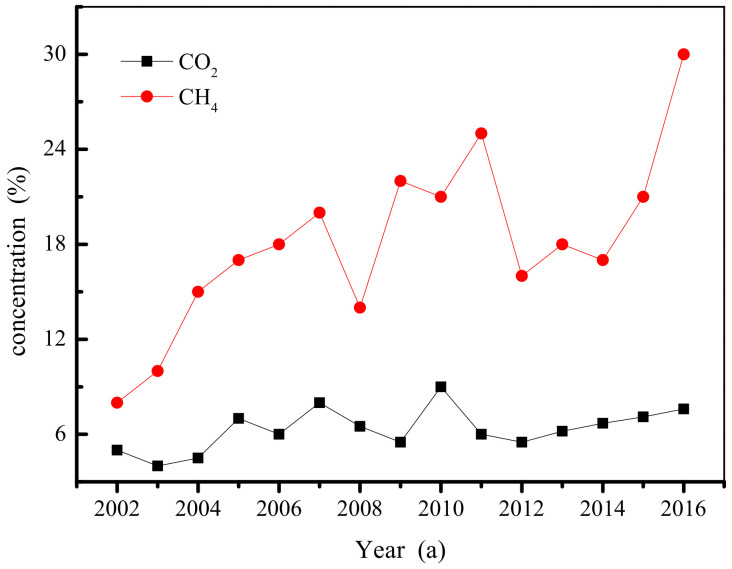
Yearly changes in the absolute gas emission of over mine.

**Figure 7 ijerph-19-04968-f007:**
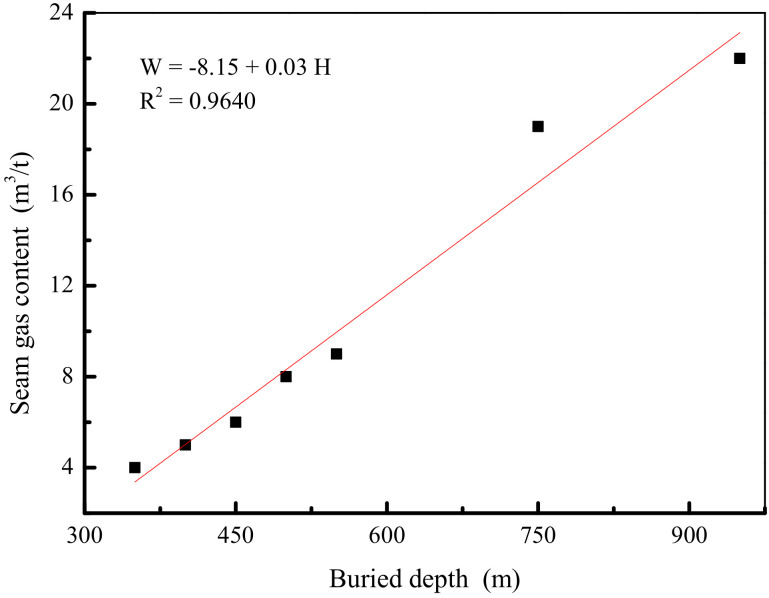
Relationship diagram of the coal gas content and depth.

**Figure 8 ijerph-19-04968-f008:**
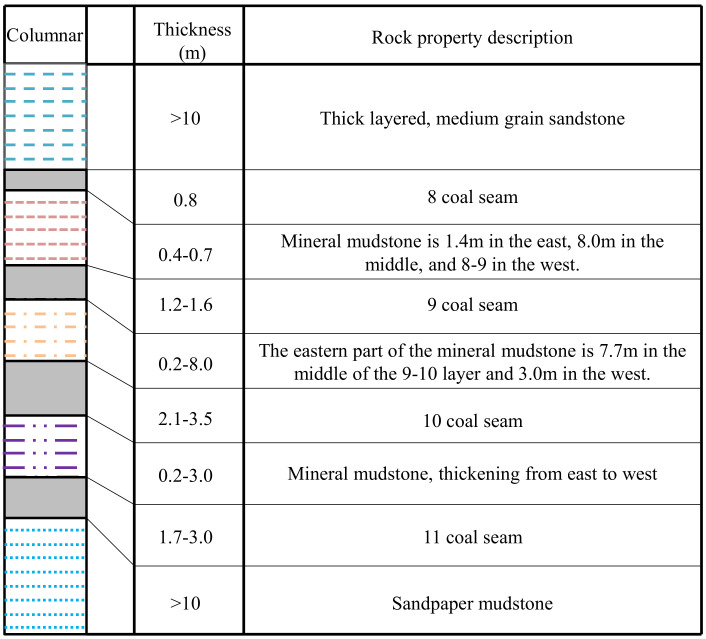
Seam histogram of group coals.

**Figure 9 ijerph-19-04968-f009:**
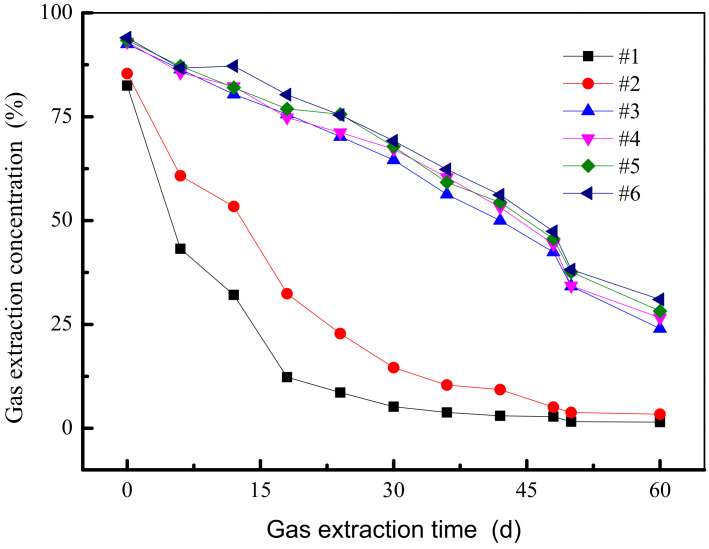
Drilling gas extraction concentration.

**Figure 10 ijerph-19-04968-f010:**
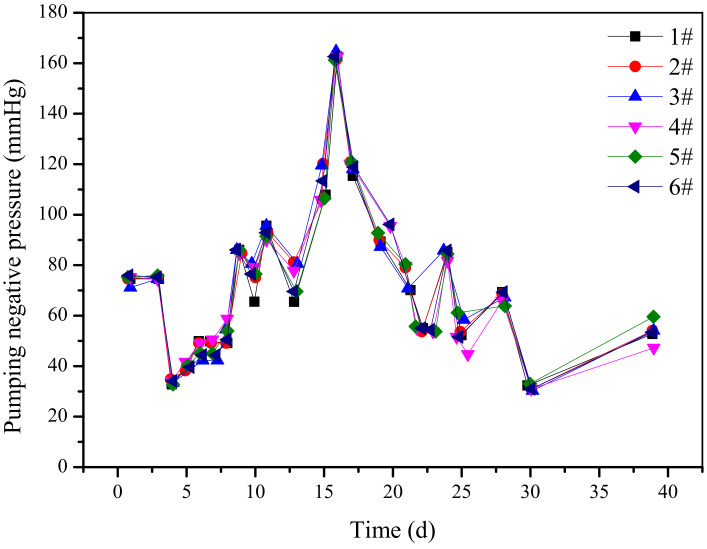
Comparison of 1#–6# borehole gas drainage pumping negative curves.

**Figure 11 ijerph-19-04968-f011:**
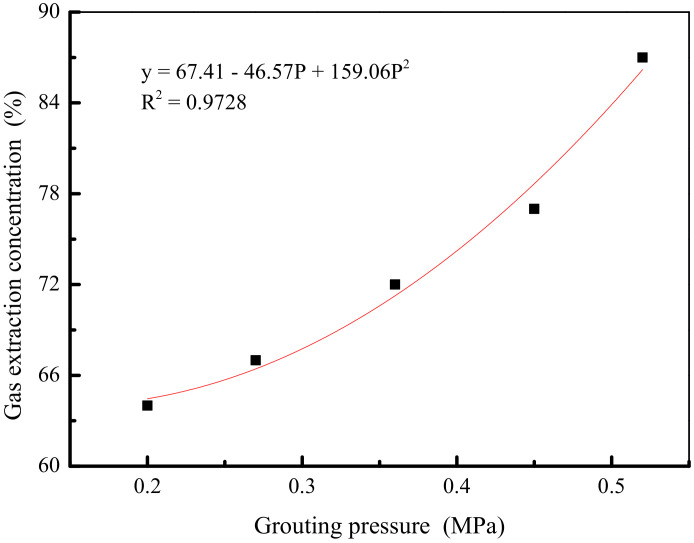
Relationship between sealing pressure and average concentration of gas extraction.

**Table 1 ijerph-19-04968-t001:** Basic parameters of time-delay foamed polyurethane.

Density		Viscosity		Mixing Ratio	Shelf Time		Casting Multiple	Dimensional Stability	Bond Strength	Compressive Strength	Curing Time
A	Β	A	Β	A:Β	A	Β	4–10 times	70 °C 48 h ≤ 2	ΜPa 3–5	ΜPa ≥12	360 s–600 s
1.17	1.42	120	60	1:1	6 months	6 months					

**Table 2 ijerph-19-04968-t002:** Basic properties of single slurry.

Water:Cement Ratio	Viscosity (s)	Specific Gravity (g/cm)	Condensation Time		Stone Body (%)	Compressive Strength (MPa)			
			Initial Setting	Final Setting		3	7	14	28
0.5:1	139	1.86	7–41	12–36	99	4.14	6.46	15.30	22.00
0.75:1	33	1.62	10–47	20–33	97	2.43	2.6	5.54	11.27
1:1	18	1.49	14–56	24–27	85	2.00	2.4	2.42	8.90
1.5:1	17	1.37	16–52	34–47	67	2.04	2.33	1.78	2.22
2:1	16	1.30	17–7	48–15	56	1.66	2.56	2.10	2.80

**Table 3 ijerph-19-04968-t003:** Proximate analysis and coal type of group coal.

Coal Seam	Type	*V_r_* (%)	H (mm)	A_g_ (%)	S (%)	Q (kcal/kg)
8	Bituminous	31.84–35	24–39 28–34	18.19–33.86 24.83–29.13	<0.4	7112–8742
9–10	Bituminous	32.63–34.97	30–34	15.83–22.55	0.31–0.63	7600–8600
11	Bituminous	31.24–32.73	22	21–37 22.79–25.92	<0.4	8562–8542

Notes: *V_r_* is volatile matter of coal without ash-based, %; H is gum layer thickness, mm; A_g_ is air drying of coal ash, %; S is sulfur of coal, %; Q is calorific value per unit mass of coal, kcal/kg.

**Table 4 ijerph-19-04968-t004:** Basic parameters determination.

*P*_0_ (MPa)	*δ* (cm)	*μ* (s)	*r*_0_ (mm)	*R* (cm)	*K* (cm/s)	*M*
0	0.04	13.5	72	60	0.005–0.08	1.28–20.44

Notes: The grouting time is generally 15 min, which is 900 s; the 525 cement can be injected into the crack width, which is generally 0.04 cm.

**Table 5 ijerph-19-04968-t005:** Gas drainage concentration of each time period of experimental drilling.

	Time	0 (d)	6 (d)	12 (d)	18 (d)	24 (d)	30 (d)	36 (d)	42 (d)	48 (d)	54 (d)	60 (d)
Number												
1#		82.5	43.2	32.1	12.3	8.6	5.2	3.8	3.0	2.8	1.6	1.5
2#		85.4	60.8	53.4	32.4	22.8	14.6	10.4	9.3	5.1	3.8	3.4
3#		92.5	86.4	80.4	75.6	70.2	64.6	56.3	50.0	42.4	34.2	24.0
4#		93.0	85.6	82.3	74.8	71.2	67.2	60.4	53.2	44.3	34.3	26.6
5#		93.4	87.2	82.0	76.9	75.6	67.8	59.2	54.4	45.6	37.6	28.2
6#		94.0	86.7	87.2	80.3	75.4	69.2	62.3	56.2	47.4	38.2	31.0

## Data Availability

The data used to support the findings of this study are available from the corresponding author upon request.

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
