# Peer review of "Determination of Key Technical Parameters in the Study of New Pressure Sealing Technology for Coal Seam Gas Extraction"

_ijerph, 2022, doi:10.3390/ijerph19094968_

Round 1

Reviewer 1 Report

Manuscript ID: ijerph_163077

Research on New Type of Pressure Sealing Technology for Gas Drainage

by

Sun, Li, Wang, Wang, Chen, Li

REFEREE’S COMMENTS

This is an interesting paper on a subject that should be of great interest to many readers. However, it is not yet suitable for publication. For location of comments, see the belows.

1.Title: It should be more concise/focused.

2.Abstract: Re-write it to present the findings of this study. Remove the introductory level information. The abstract should be supported by some quantitative findings.

3.Introduction: Literature review should be strengthened. Last paragraph; the authors should clearly indicate the originality/novelty of their study. Current problems with sealing technology: This section could be combined with the Introduction section.

4.Experimental study on pressure sealing borehole in coal seam: Which equipment did the authors employed?

5.Re-write the caption of Figure 1. Caption should be at the bottom of Figure.

6.How did the authors derive the equations 6 and 7?

7.Field Application: Figure 3 is not clearly seen. page 8-13; cite the relevant papers. Discussion: page 15-16; cite the relevant papers.

8.Conclusions:The authors should support their findings using quantitatively.

9.In General: Language of the text was found to be fine. Check out the details of the references cited. Recently published papers should be cited through the text.

Best regards,

Author Response

  1. Comment: 1.Title: It should be more concise/focused.

Response: Based on your comments, we've reformulated the title of the manuscript to make it more concise/focused, and the changes will be marked in red in the manuscript. This has greatly helped to improve the comprehensiveness and international readability of the introduction of this manuscript.

  1. Comment: 2.Abstract: Re-write it to present the findings of this study. Remove the introductory level information. The abstract should be supported by some quantitative findings.

Response: Based on your comments, we have re-edited the abstract, removed the introductory level information, and introduced some support from the quantitative findings, which are marked in red in the manuscript, and hope that the changes will satisfy your comments.

  1. Comment: Introduction: Literature review should be strengthened. Last paragraph; the authors should clearly indicate the originality/novelty of their study. Current problems with sealing technology: This section could be combined with the Introduction section.

Response: We sincerely agree with your comments. Based on your comments we have re-edited the Introduction section to strengthen the literature review, clearly indicating the originality/novelty of the research and marked in red in the manuscript, hoping that the changes will satisfy your comment request.

  1. Comment: Experimental study on pressure sealing borehole in coal seam: Which equipment did the authors employed?

Response: Equipment used in the coal seam pressure sealing test Main pressure gauges, grouting pumps and other instruments and equipment

  1. Comment: Re-write the caption of Figure 1. Caption should be at the bottom of Figure.

Response: Based on your comments, we have rewritten the title of Figure 1 and placed the title at the bottom of the figure.

  1. Comment: How did the authors derive the equations 6 and 7?

Response: In order to ensure the grouting effect, the manuscript adopts both the theoretical model and some experience in on-site coal seam grouting technology, and the empirical theoretical model formula is shifted and deformed to obtain the grouting pressure calculation formula required in this manuscript.

  1. Comment: Field Application: Figure 3 is not clearly seen. page 8-13; cite the relevant papers. Discussion: page 15-16; cite the relevant papers.

Response: According to your bottle mouth, we have changed the content of Figure 3 to make the picture more clear. Added relevant cited papers to pages 8-13 of the manuscript. Added citations to related papers on pages 15-16. Modified content in the text is marked in red.

  1. Comment: Conclusions:The authors should support their findings using quantitatively.

Response: Based on your comments, we have revised the conclusions and added quantitative data to support the innovations of this paper.

  1. Comment: Check out the details of the references cited. Recently published papers should be cited through the text.

Response: Based on your comments, we have re-examined the references section of the paper and made changes to those that do not meet the main text citation requirements, and added citations to recently published papers. Your comment is extremely important in improving our manuscript, Thank you again.

Reviewer 2 Report

Highlight changes in yellow in a next revision, please. No track changes.

I understand the manuscript is aimed at:

https://www.mdpi.com/journal/ijerph/special_issues/Safety_Management_of_Coal_and_Rock_Dynamic_Disasters

Accordingly, the Keywords suggested in the call should be better addressed in the manuscript to adequate the Journal, a serious publication factor being discussed in the related media

Abstract: I could not fully distinguish the contextualization, from the methodology used in this manuscript, as real findings and practical implications… What is the exact purpose?

The introductions should end with the clear objectives

Three references are used here: “2. Current problems with sealing technology

A significant part has no refrences:

“Among them, cement mortar sealing is mainly suitable for

inclined drilling, which is not suitable for near horizontal or gently inclined coal seam;

foamed polymeric material sealing has the advantages of high foaming ratio, light weight

and rapid sealing of polymeric materials, but its sealing The material cost of the borehole

is high and the operation requirement is high; the fast sealing device has a high sealing

speed, and the repeated use can reduce the cost, but the effect is poor, and it is only suitable for temporary drainage and sealing; in general, the existing gas drainage method only It is limited to how to “block” the upper borehole, but does not involve how to block and

deal with coal seam fissures and gas leakage channels. These coal seam fissures and gas

leakage channels will develop and expand with the extraction of gas, resulting in gas potential in the later stage of pumping. It is released, causing deformation, displacement and pressure relief of the coal body. Under the action of suction negative pressure, air can

easily enter the borehole through these cracks, resulting in a decrease in gas drainage concentration and shortening the drainage cycle of gas drilling.”

Accordingly, these two theoretical introduction parts are scarcely referenced…

Revise headings to be self-explanatory:

3.1. Technical principle

3.2. Technical procedure

3.3. Technical parameter determination

Avoid repeating terms…

Table 1. Basic parameters of delay polyurethane.”

Table 2. Basic properties of single slurry.”

Table 4. Basic parameters determination.”

Says nothing…

Repetition is not good in a scientific text… at all…

After, not before the image:

Figure 2. Schematic diagram of polyurethane and slurry sealing.”

Address italics and include units inside (), wherever available… “Where, L1”

Assure thar every known equations has the reference immediately before EACH one is presented…

It seems not to be the case

Check also the format and distortion

Address the italics in parameters inside figures…

Figure 5. Relationship diagram of the coal gas content and the vertical depth.”

I do not understand the captions…

Not enlightening. Check papers from other highly cited authors too

Use “notes”, check italics… “Where, Vr is combustible body volatiles; H is gum layer thickness; Ag is ash; S is sulfur; Q is calorific value.”

Also in table’s headings…

?! “Figure 6. seam histogram of Group coals. Figure or table?”

Please avoid the “listing” headings all over... “(1) Sealing length”

This is not a thesis…

“as shown in Fig. 9. Fig.9 reflects” revise duplication in language

I personally do not appreciate all theses duplications and “listing style”. The authors should be able to merge the content, resulting in a relevant results and discussion section, much more difficult to accomplish:

5. Discussion

5.1. Influencing factor

5.1.1. Coal body characteristics”

Theis makes no sense in a discussion, bringing new content:

Figure 10. Diagram of coal pore type.”

Figure 11. Schematic diagram of seepage in the crack with grout.”

To be moved to the introductory sections, this is not a new literature section…

“Specific effects of cracks in coal body on pressure sealing technology [40-45] :

  • Influence of coal body crack width”

And goes on and on

If merged with the results sections, authors will be forced to significantly shorten the content

Conclusions:

Please include brief contextualization and methodology, main findings and practical implications, aiming to answered to the objectives at the beginning

Please, do not… “list”:

“The main conclusions are as follows:”

Highlight novelty and innovation

I would advise the authors to significantly further work on the relevance of this manuscript, particularly if aimed to this specific journal

Extensive restructuring needs to be carried out, specifically in the discussion.

References:

Mostly centred in Chinese sources, which is not the aim in an international indexed journal

Include 2022 references

Author Response

  1. Comment: Abstract: I could not fully distinguish the contextualization, from the methodology used in this manuscript, as real findings and practical implications… What is the exact purpose?

Response: Based on your comments, we have re-edited the abstract, removed the introductory level information, and introduced some support from the quantitative findings, which are marked in red in the manuscript, and hope that the changes will satisfy your comments.

  1. Comment: The introductions should end with the clear objectives.

Response: We sincerely agree with your opinion. Based on your comments, we have re-edited the Introduction section to end with a clear goal, clearly indicating the originality/novelty of the study, and marked in red in the manuscript, hoping that the changes have been made to meet your request for comment.

  1. Comment: Three references are used here: “2. Current problems with sealing technology” A significant part has no refrences: “Among them, cement mortar sealing is mainly suitable for inclined drilling, which is not suitable for near horizontal or gently inclined coal seam; foamed polymeric material sealing has the advantages of high foaming ratio, light weight and rapid sealing of polymeric materials, but its sealing The material cost of the borehole is high and the operation requirement is high; the fast sealing device has a high sealing speed, and the repeated use can reduce the cost, but the effect is poor, and it is only suitable for temporary drainage and sealing; in general, the existing gas drainage method only It is limited to how to “block” the upper borehole, but does not involve how to block and deal with coal seam fissures and gas leakage channels. These coal seam fissures and gas leakage channels will develop and expand with the extraction of gas, resulting in gas potential in the later stage of pumping. It is released, causing deformation, displacement and pressure relief of the coal body. Under the action of suction negative pressure, air can easily enter the borehole through these cracks, resulting in a decrease in gas drainage concentration and shortening the drainage cycle of gas drilling.”

Accordingly, these two theoretical introduction parts are scarcely referenced….

Response: We sincerely agree with your opinion. Based on your comments, we have added reference citations, marked in red in the manuscript. Although these two theories are rarely cited in the introduction, they can play an important role in explaining the current problems faced by was drainage boreholes, and are therefore presented in this manuscript. Your comment is extremely important in improving our manuscript, Thank you again.

  1. Comment: Revise headings to be self-explanatory:

“3.1. Technical principle”

“3.2. Technical procedure”

“3.3. Technical parameter determination”

Avoid repeating terms…

“Table 1. Basic parameters of delay polyurethane.”

“Table 2. Basic properties of single slurry.”

“Table 4. Basic parameters determination.”

Says nothing…

Repetition is not good in a scientific text… at all…

Response: We sincerely agree with your opinion. Based on your comments, we have revised the manuscript titled Self-explanatory and marked the revisions in red. The text has been re-edited to omit repeated terms in the manuscript, and table titles have been revised to reflect the scientific text.

  1. Comment: After, not before the image:“Figure 2. Schematic diagram of polyurethane and slurry sealing.”

Response: Based on your comments, we have rewritten the title of Figure 2 and placed the title at the bottom of the figure.

  1. Comment: Address italics and include units inside (), wherever available… “Where, L1”

Response: We sincerely agree with your opinion. Based on your comments, we have italicized addresses in the manuscript and added units in ( ), and marked the revisions in red.

  1. Comment: Assure thar every known equations has the reference immediately before EACH one is presented…

It seems not to be the case

Check also the format and distortion

Response: We sincerely agree with your opinion. Based on your comments, we checked the manuscript throughout to ensure that every known equation was referenced before each known equation appeared, and revised the manuscript content formatting and distortions to ensure that the manuscript format meets journal requirements.

  1. Comment: Address the italics in parameters inside figures…

“Figure 5. Relationship diagram of the coal gas content and the vertical depth.”

I do not understand the captions…

Not enlightening. Check papers from other highly cited authors too

 Use “notes”, check italics… “Where, Vr is combustible body volatiles; H is gum layer thickness; Ag is ash; S is sulfur; Q is calorific value.”

Also in table’s headings…

?! “Figure 6. seam histogram of Group coals. Figure or table?”

Address the italics in parameters inside figures…

“Figure 5. Relationship diagram of the coal gas content and the vertical depth.”

I do not understand the captions…

Not enlightening. Check papers from other highly cited authors too

Response: We sincerely agree with your opinion. Based on your comments, we checked the manuscript throughout to ensure that every known equation was referenced before each known equation appeared, and revised the manuscript content formatting and distortions to ensure that the manuscript format meets journal requirements.

  1. Comment: Use “notes”, check italics… “Where, Vr is combustible body volatiles; H is gum layer thickness; Ag is ash; S is sulfur; Q is calorific value.”

Also in table’s headings…

?! “Figure 6. seam histogram of Group coals. Figure or table?” 

Please avoid the “listing” headings all over... “(1) Sealing length”

This is not a thesis… 

“as shown in Fig. 9. Fig.9 reflects” revise duplication in language

Response: We sincerely agree with your opinion. Based on your comments, we have reviewed the manuscript for use of "notes", italics checks, title checks, and revised manuscripts for errors. The coal seam histogram of coal formation is definitely a graph, not a table. The "List" headings appearing in the manuscript have been revised based on comments. Modified Figure 9 to reflect repeated language.

  1. Comment: I personally do not appreciate all theses duplications and “listing style”. The authors should be able to merge the content, resulting in a relevant results and discussion section, much more difficult to accomplish:

“5. Discussion 5.1. Influencing factor 5.1.1. Coal body characteristics”

Response: We sincerely agree with your opinion. Based on your comments, we have optimized the incorporation of what can be incorporated in the manuscript, resulting in a relevant Results and Discussion section, which is critical to improving the quality of our manuscript, thank you very much.

  1. Comment: Theis makes no sense in a discussion, bringing new content:

“Figure 10. Diagram of coal pore type.” “Figure 11. Schematic diagram of seepage in the crack with grout.” To be moved to the introductory sections, this is not a new literature section…Specific effects of cracks in coal body on pressure sealing technology [40-45] :ï‚·Influence of coal body crack width” And goes on and on

If merged with the results sections, authors will be forced to significantly shorten the content

Response: We sincerely agree with your opinion. Based on your comments, we have moved "Figure 10. Coal Pore Type Map." in the manuscript to the Introduction section and have made significant revisions to the Introduction section, which have contributed to a significant improvement in the quality of the manuscript.

  1. Comment: Conclusions: Please include brief contextualization and methodology, main findings and practical implications, aiming to answered to the objectives at the beginning

Please, do not… “list”:

“The main conclusions are as follows:” 

Highlight novelty and innovation 

I would advise the authors to significantly further work on the relevance of this manuscript, particularly if aimed to this specific journal Extensive restructuring needs to be carried out, specifically in the discussion.

Response: We sincerely agree with your opinion. Based on your comments, we have re-edited the content of the conclusion, in order to highlight the novelty and innovation of the conclusion, as well as the relevance of the manuscript content for further discussion, which is marked in red in the manuscript.

  1. Comment: References: Mostly centred in Chinese sources, which is not the aim in an international indexed journal

Include 2022 references

Response: We sincerely agree with your opinion. Based on your comments, we have re-cited editorial references, with the goal of indexing journals internationally to avoid Chinese-language sources as the center, heavily citing 2022 references, and treating them in red in the manuscript. Your comment is extremely important in improving our manuscript, Thank you again.

Reviewer 3 Report

The article refers to increasing the recovery of methane gas stored in coal seams through a new type of pressure sealing technology for gas drainage. It is a very interesting subject, too little researched worldwide, because in the case of most coal mines this gas is released into the atmosphere. The existing methane gas in coal deposits in various forms, following the exploitation of coal, is released into underground mining workings, from where it is discharged into the atmosphere with the help of ventilation installations. Methane gas has an impact on the environment, in terms of the greenhouse effect, about 20-25 times higher than carbon dioxide. Therefore, if it is captured before the exploitation of the deposit and used in combustion plants, this effect is reduced accordingly. Moreover, its evacuation from the deposits is very important to avoid methane explosions, which endanger the lives of workers and the condition of the equipment. The use of methane gas in combustion plants is limited due to the concentration of exhaust gas. Proper sealing of boreholes can restore methane gas to a concentration and quantity that makes the use of methane gas in combustion plants both technically and economically efficient. The proposed technology is based on the existence around the underground excavations of the cracks generated by the concentration of stresses, implicitly transmitted around the boreholes, which determines the increase of the necessary sealing section of the boreholes. In this sense it is proposed to use the pressure grouting to block the gas leakage micropores and crack channels and so that the slurry solidifies and has a higher viscosity to the coal body. Also, the organic plastomer improves the coal seams fissures and the integrity and homogeneity of the coal seams. The synthesis of the research presented in the article is well argued and substantiated both theoretically and practically, with certain deficiencies related to the expression in English. Following the publication of this article, the specialized literature of the researched field is improved, with an impact on solving a problem of both occupational safety and environmental pollution. Therefore, I propose to publish this article with an in-depth review of the English language.

Author Response

Thank you for your letter and for the reviewers’ comments concerning our manuscript entitled “Research on New Type of Pressure Sealing Technology for Gas Drainage in Coal Seam”. Those comments are all valuable and very helpful for revising and improving our paper, as well as the important guiding significance to our researches. We have studied comments carefully and have made correction which we hope meet with approval. Revised portion are marked in red in the paper. Please check it.

Round 2

Reviewer 1 Report

-

Author Response

Dear Editor:

Thank you for your letter and for the reviewers’ comments concerning our manuscript entitled “Research on New Type of Pressure Sealing Technology for Gas Drainage in Coal Seam”. Those comments are all valuable and very helpful for revising and improving our paper, as well as the important guiding significance to our researches. We have studied comments carefully and have made correction which we hope meet with approval. Revised portion are marked in red in the paper. The main corrections in the paper and the responds to the reviewer’s comments are as flowing.

Responds to the reviewers’ comments:

  1. Comment: Highlight changes in yellow in a next revision, please. No track changes. 

Please, DO NOT send tracked changed documents. Extremely difficult to review

I need a clear document. I need a clean version with changes highlighted. 

It does not, where are the objectives of this paper?

  1. Comment: The introductions should end with the clear objectives.

Response: We sincerely agree with your opinion. Based on your comments, we have re-edited the Introduction section to end with a clear goal, clearly indicating the originality/novelty of the study, and marked in red in the manuscript, hoping that the changes have been made to meet your request for comment. ”

  1. Comment: The process…, the process… avoid duplications. 

I was extensive before, so I will not do it again. I would advise the authors to go back to see the comments and make the manuscript relevant

Namely addressing headings mentioned in the previous revision and a better connection, and not separations, between the discussion and results sections.

 This is not good:

“5.1.3. Grouting program

(1) Influence of injection method” 

Authors did not significantly address the conclusions, namely contextualization 

Please, do not… “list”

“The main conclusions are as follows:”

No answer to all, to mu initial comment:

“I understand the manuscript is aimed at:

https://www.mdpi.com/journal/ijerph/special_issues/Safety_Management_of_Coal_and_Rock_Dynamic_Disasters

Accordingly, the Keywords suggested in the call should be better addressed in the manuscript to adequate the Journal, a serious publication factor being discussed in the related media”.

Response: Based on your comments, we've reformulated the title of the manuscript to make it more concise/focused, and the changes will be marked in yellow in the manuscript. This has greatly helped to improve the comprehensiveness and international readability of the introduction of this manuscript.

Reviewer 2 Report

Highlight changes in yellow in a next revision, please. No track changes.

Please, DO NOT send tracked changed documents. Extremely difficult to review

I need a clear document. I need a clean version with changes highlighted.

It does not, where are the objectives of this paper?

2. Comment: The introductions should end with the clear objectives.

Response: We sincerely agree with your opinion. Based on your comments, we have re-edited the Introduction section to end with a clear goal, clearly indicating the originality/novelty of the study, and marked in red in the manuscript, hoping that the changes have been made to meet your request for comment. ”

The process…, the process… avoid duplications.

I was extensive before, so I will not do it again. I would advise the authors to go back to see the comments and make the manuscript relevant

Namely addressing headings mentioned in the previous revision and a better connection, and not separations, between the discussion and results sections.

This is not good:

“5.1.3. Grouting program
(1) Influence of injection method”

Authors did not significantly address the conclusions, namely contextualization

Please, do not… “list”

“The main conclusions are as follows:”

No answer to all, to mu initial comment:

“I understand the manuscript is aimed at:

https://www.mdpi.com/journal/ijerph/special_issues/Safety_Management_of_Coal_and_Rock_Dynamic_Disasters

Accordingly, the Keywords suggested in the call should be better addressed in the manuscript to adequate the Journal, a serious publication factor being discussed in the related media”

Author Response

Dear Editor:

Thank you for your letter and for the reviewers’ comments concerning our manuscript entitled “Research on New Type of Pressure Sealing Technology for Gas Drainage in Coal Seam”. Those comments are all valuable and very helpful for revising and improving our paper, as well as the important guiding significance to our researches. We have studied comments carefully and have made correction which we hope meet with approval. Revised portion are marked in red in the paper. The main corrections in the paper and the responds to the reviewer’s comments are as flowing.

Responds to the reviewers’ comments:

  1. Comment: Highlight changes in yellow in a next revision, please. No track changes.

 Why is this new? Clarify it through the manuscript:

“Determination of key technical parameters in the study of new
pressure sealing technology for coal seam gas extraction”

The term NEW is found a few times (not in the abstract)

Novelty/originality/innovation is unclear. Never found

I cannot see any real quantitative findings:

“Author’s response:

Comment: Abstract: I could not fully distinguish the contextualization, from the methodology used in this manuscript, as real findings and practical implications… What is the exact purpose?

Response: Based on your comments, we have re-edited the abstract, removed the introductory level information, and introduced some support from the quantitative findings, which are marked in red in the manuscript, and hope that the changes will satisfy your comments.”

 I do not understand the meaning of this sentence nor its contextualization:

“The gas release crack is the principle of the pressure-sealing technology of the coal
seam gas drainage belt, and the pressure-sealing technology suitable for the coal seam is designed.”

 Do not list:

“Field application practice has proved that:”

Where are the real quantitative findings?

This says nothing:

“is far lower than the sealing o” ?!

“â‘£ Sealing with pressure The hole process is reliable and stable.”

Clearly, the manuscript further revision…

I was extensive before; I will not repeat myself. I can see authors ignored previous extensive comments

I suggest to go through them again…

Duplication of expressions, objectives at the end of introduction, etc:

The process.. the process…

“2. Comment: The introductions should end with the clear objectives.

Response: We sincerely agree with your opinion. Based on your comments, we have re-edited the Introduction section to end with a clear goal, clearly indicating the originality/novelty of the study, and marked in red in the manuscript, hoping that the changes have been made to meet your request for comment. ”

Everything you say, not resulting from your study needs references, where are they?

“These coal seam fissures
and gas leakage channels will develop and expand with the extraction of gas, resulting in
gas potential in the later stage of pumping. It is released, causing deformation, displacement and pressure relief of the coal body. Under the action of suction negative pressure,
air can easily enter the borehole through these cracks, resulting in a decrease in gas drainage concentration and shortening the drainage cycle of gas drilling.”

Again, says nothing:

“Table 2. Basic properties of single slurry.”

Nor I understand the English nor do I consider these are equations…:

“Which is:
r r mm 1 2 < +10(4)
(5)

r mm r 3 4 + < 10”

 No formulas, equations!

“When applying, the rationality of the grouting pressure is checked by the formula (7)
and the formula (8), and the formula (9) is used as the on-site grouting pressure.”

 Captions need strong revision, again:

“Fig 5, 7, etc”

Check italics in parameters, again!

I do not understand the connection between the headings:

“3.2. Technical parameter determination
3.2.1. Polyurethane”

And then..

“4.2. Experimental parameter determination”?!

Compare to in a different section:

““3.2. Technical parameter determination”

 Headings in the table must be present and clear:

“Table 5. Gas drainage concentration of each time period of experimental drilling”
What is in the first column and row?!

 This needs to be changed to be enlightening. Check other +papers and do not repeat beginning of captions:

“Figure 11. Relationship”

This is not a thesis, do not use continuous headings without any text connection…

“5. Discussion
5.1. Influencing factor
5.1.1. Coal characteristics”

Unclear, avoid listing:

“Specific effects of cracks in coal body on pressure sealing technology [40-45] :”

Why should this be considered discussion?

“Figure 12. Schematic diagram of seepage in the crack with grout.”

Please do not list, easy to write, difficult to connect:

“The main conclusions are as follows:”

Where are the quantitative results, , implications, innovation and novelty?

“(3) It can be seen from the field test that the process meets the technical requirements
of the gas drainage and drilling engineering of the coal seam, and provides a new scientific
and effective sealing method for the direct extraction and utilization of the coal seam gas.
It is worth promoting.”

I do feel the changes made are not enough:

“We tried our best to improve the manuscript and made some changes in the manuscript”

 Most references addictions, are, again, from the same region…

Again:

“I would advise the authors to significantly further work on the relevance of this manuscript, particularly if aimed to this specific journal Extensive restructuring needs to be carried out, specifically in the discussion.”  

It is my opinion that the manuscript lacks strong improvements. Addressed in earlier revisions

Response: Based on your comments, we have re-edited the manuscript, revised the manuscript title, changed the Introduction, Abstract, and Conclusion, and made changes to the text to make the manuscript more concise/focused, and the changes will be marked in yellow in the manuscript. Some of the tables presented in the text support the content of the manuscript and greatly contribute to the comprehensiveness and international readability of the introduction to this manuscript.

Round 3

Reviewer 2 Report

Highlight changes in yellow in a next revision, please. No track changes.

Why is this new? Clarify it through the manuscript:

Determination of key technical parameters in the study of new
pressure sealing technology for coal seam gas extraction

The term NEW is found a few times (not in the abstract)

Novelty/originality/innovation is unclear. Never found

I cannot see any real quantitative findings:

“Author’s response:

  1. Comment: Abstract: I could not fully distinguish the contextualization, from the methodology used in this manuscript, as real findings and practical implications… What is the exact purpose?

Response: Based on your comments, we have re-edited the abstract, removed the introductory level information, and introduced some support from the quantitative findings, which are marked in red in the manuscript, and hope that the changes will satisfy your comments.”

I do not understand the meaning of this sentence nor its contextualization:

“The gas release crack is the principle of the pressure-sealing technology of the coal
seam gas drainage belt, and the pressure-sealing technology suitable for the coal seam is designed.”

Do not list:

“Field application practice has proved that:”

Where are the real quantitative findings?

This says nothing:

“is far lower than the sealing o”

?!

“â‘£ Sealing with pressure The hole process is reliable and stable.”

Clearly, the manuscript further revision…

I was extensive before; I will not repeat myself. I can see authors ignored previous extensive comments

I suggest to go through them again…

Duplication of expressions, objectives at the end of introduction, etc:

The process.. the process…

2. Comment: The introductions should end with the clear objectives.

Response: We sincerely agree with your opinion. Based on your comments, we have re-edited the Introduction section to end with a clear goal, clearly indicating the originality/novelty of the study, and marked in red in the manuscript, hoping that the changes have been made to meet your request for comment. ”

Everything you say, not resulting from your study needs references, where are they?

“These coal seam fissures
and gas leakage channels will develop and expand with the extraction of gas, resulting in
gas potential in the later stage of pumping. It is released, causing deformation, displacement and pressure relief of the coal body. Under the action of suction negative pressure,
air can easily enter the borehole through these cracks, resulting in a decrease in gas drainage concentration and shortening the drainage cycle of gas drilling.”

Again, says nothing:

Table 2. Basic properties of single slurry.”

Nor I understand the English nor do I consider these are equations…:

“Which is:
r r mm 1 2 < +10

(4)
(5)

r mm r 3 4 + < 10”

No formulas, equations!

“When applying, the rationality of the grouting pressure is checked by the formula (7)
and the formula (8), and the formula (9) is used as the on-site grouting pressure.”

Captions need strong revision, again:

“Fig 5, 7, etc”

Check italics in parameters, again!

I do not understand the connection between the headings:

3.2. Technical parameter determination
3.2.1. Polyurethane”

And then..

4.2. Experimental parameter determination”?!

Compare to in a different section:

““3.2. Technical parameter determination

Headings in the table must be present and clear:

Table 5. Gas drainage concentration of each time period of experimental drilling”
What is in the first column and row?!

This needs to be changed to be enlightening. Check other +papers and do not repeat beginning of captions:

“Figure 11. Relationship”

This is not a thesis, do not use continuous headings without any text connection…

5. Discussion
5.1. Influencing factor
5.1.1. Coal characteristics”

Unclear, avoid listing:

“Specific effects of cracks in coal body on pressure sealing technology [40-45] :”

Why should this be considered discussion?

Figure 12. Schematic diagram of seepage in the crack with grout.”

Please do not list, easy to write, difficult to connect:

“The main conclusions are as follows:”

Where are the quantitative results, , implications, innovation and novelty?

“(3) It can be seen from the field test that the process meets the technical requirements
of the gas drainage and drilling engineering of the coal seam, and provides a new scientific
and effective sealing method for the direct extraction and utilization of the coal seam gas.
It is worth promoting.”

I do feel the changes made are not enough:

“We tried our best to improve the manuscript and made some changes in the manuscript”

Most references addictions, are, again, from the same region…

Again:

“I would advise the authors to significantly further work on the relevance of this manuscript, particularly if aimed to this specific journal Extensive restructuring needs to be carried out, specifically in the discussion.”

It is my opinion that the manuscript lacks strong improvements. Addressed in earlier revisions

Author Response

(The authors gave the same response as above.)
